# An Assessment of Relations between Vegetation Green FPAR and Vegetation Indices through a Radiative Transfer Model

**DOI:** 10.3390/plants12101927

**Published:** 2023-05-09

**Authors:** Shouzhen Liang, Wandong Ma, Xueyan Sui, Meng Wang, Hongzhong Li

**Affiliations:** 1Shandong Academy of Agricultural Sciences, Ji’nan 250100, China; 2Satellite Environment Center for Ecology and Environment, Ministry of Ecology and Environment, Beijing 100094, China; 3Shenzhen Institutes of Advanced Technology, Chinese Academy of Sciences, Shenzhen 518055, China

**Keywords:** green FPAR, SAIL, forest, vegetation index

## Abstract

The fraction of absorbed photosynthetically active radiation (FPAR) is widely used in remote sensing-based production models to estimate gross or net primary production. The forest canopy is composed primarily of photosynthetically active vegetation (PAV, green leaves) and non-photosynthetic vegetation (NPV e.g., branches), which absorb PAR but only the PAR absorbed by PAV is used for photosynthesis. Green FPAR (the fraction of PAR absorbed by PAV) is essential for the accurate estimation of GPP. In this study, the scattering by arbitrary inclined leaves (SAIL) model was reconfigured to partition the PAR absorbed by forest canopies. The characteristics of green FPAR and its relationships with spectral vegetation indices (NDVI, EVI, EVI2, and SAVI) were analyzed. The results showed that green FPAR varied with the canopy structure. In the forests with high coverage, the green FPAR was close to the total FPAR, while in the open forests, the green FPAR was far smaller than the total FPAR. Plant area index had more important impacts on the green FPAR than the proportion of PAV and optical properties of PAV. The significant relationships were found between spectral vegetation indices and the green FPAR, but EVI was more suitable to describe the variation of canopy green FPAR.

## 1. Introduction

Photosynthetically active radiation (PAR) refers to the radiation with the spectral range from 400–700 nm that is used by plants in photosynthesis. The fraction of absorbed photosynthetically active radiation (FPAR) is the ratio of the PAR absorbed by vegetation to the PAR across an integrated plant canopy [1]. FPAR is closely linked to canopy functioning processes such as canopy photosynthesis, carbon assimilation and evapotranspiration rates. It is a key biophysical variable for vegetation productivity estimation, vegetation growth condition monitoring, and climate change analysis [2]. An accurate specification of FPAR is an important detail in large scale productivity and carbon budget models [3]. FPAR is recognized by the global climate observing system (GCOS) as essential climate variables (ECVs) [4].

Traditionally, the canopy total FPAR is directly measured in the field through optical instruments including hemispherical photographs [5], the tracking radiation and architecture of canopies (TRAC) instrument [6], the burr Brown data acquisition system (BBDAS) [7]. Traditional method is time-consuming and expensive. This makes assessment of FAPR on the landscape and ecosystem scales impractical. As an alternative solution, FPAR can be derived through remote measures of surface spectral reflectance, and remote sensing-based methods are currently the only feasible way of acquiring FPAR estimates at the temporal and spatial scales [8,9,10,11]. A number of state of the art algorithms have been proposed to estimate the important environmental variable. They can be categorized as, (a) empirical methods based on relationships between FPAR and vegetation indices, LAI (Leaf area index); (b) physical methods based on the physics of radiation interaction with elements of a canopy and transport within the vegetative medium; (c) machine learning algorithms [12,13]. Moreover, the number of researches utilizing hybrid regression methods combining radiative transfer model simulations with machine learning regression methods to derive vegetation FPAR is increasing [14]. During the recent decades, a series of satellite FPAR products have been developed based on different definitions, assumptions, retrieval algorithms, and sensors including MODerate-resolution Imaging Spectroradiometer (MODIS), Multi-angle Imaging SpectroRadiometer (MISR), MEdium Resolution Imaging Spectrometer (MERIS), Sea-Viewing Wide Field-of-View Sensor (SeaWiFS), GEOV1,Global Land Surface Satellite (GLASS), CYCLOPES, Visible Infrared Imaging Radiometer (VIIRS), FPAR3 g, Copernicus Global Land Service (CGLS), Earth Polychromatic Imaging Camera (EPIC) and Sentinel-3 Ocean and Land Colour Instrument (OLCI) [13,15].

For woody plants, the canopy is composed of photosynthetically active vegetation (PAV, mostly green leaves) and non-photosynthetically active vegetation (NPV, e.g., braches, stems) [16]. PAV and NPV absorb PAR, but only the PAR absorbed by PAV is used for photosynthesis [17,18]. Obviously, this quantity of FPAR determined by PAV is lower than the canopy total FPAR because it does not include PAR absorption by wood materials. For forest ecosystems, NPV can increase canopy FPAR by 10–40% when leaf area index is less than 3.0 [8]. If the NPV contribution to canopy FPAR is not removed, the PAR used to photosynthesis will be overestimated [19]. Additionally, the total FPAR cannot accurately reflect the spatiotemporal variations in photosynthesis due to the varying fractions of NPV [20,21]. Therefore, for the canopy consist of PAV and NPV, the FPAR should be partitioned into FPAR_PAV_ (green FPAR) and FPAR_NPV_, and FPAR_PAV_ should be estimated instead of the canopy total FPAR in order to improve the estimation accuracy of vegetation productivity. Any model that accounts for FAPA_RPAV_ is likely to substantially improve estimation of GPP or NPP of forests, given a known value of light-use efficiency [22]. However, there is no practical direct method of separating the PAR absorbed by the green leaves [23,24]. In some researches, green FPAR is defined as the product of the total FPAR and the fraction of leaf area in the canopy [2,18]. That approach is based on the idea that the green leaves and NPV have the same spectral characteristics. For forest canopies, NPV is significantly different from green leaves, which makes the approach invalid. Fortunately, green FPAR can be calculated from radiative transfer models describing the transfer of solar radiation in plant canopies [25,26]. Physics-based radiative transfer models represent the scattering and absorption of radiation by scattering elements of canopies. These models are helpful for scientists to understand how radiations interact with the environment, and how they propagate towards the sensor. But such an approach calculating FPAR through radiative transfer models is time consuming [20]. Vegetation indices play key roles in FPAR inversion based on remotely sensed data. Live green plants strongly absorb solar radiation in red spectral region, and scatter (reflect and transmit) solar radiation in the near-infrared spectral region. Vegetation indices derived from radiometric measurements in red and near-infrared wavelengths can reflect absorption and reflection of vegetation to solar radiation. Consequently, some vegetation indices (e.g., NDVI, EVI) are related to FPAR, and are frequently used to estimated FPAR due to its ease of use and simplicity [13,26,27,28,29]. Relationships between vegetation indices and FPAR_PAV_ need be assessed for forest ecosystems in order to inverse the green FPAR_PAV_.

In this study, the scattering by arbitrary inclined leaves model (SAIL model) was used to partition the canopy FPAR. The aims of our study are twofold, (I) to simulate and analyze the variation of green FPAR with the canopy structure; (II) to explore the relations between the green FPAR and vegetation indices for estimation of green FPAR.

## 2. Results and Discussion

### 2.1. The Variations of FPAR and Vegetation Indices

FPAR_PAV_ (green FPAR), FPAR_NPV_ and the total FPAR of deciduous broadleaf forests demonstrated different modes with canopy structure variations (Figure 1). The total FPAR and green FPAR had similar curves with plant area index. They increased as the canopy plant area index was rising. FPAR_NPV_ demonstrated different trajectories from the total FPAR and green FPAR. FPAR_NPV_ had maximum values when PAI was 2.0. As PAI was below 2.0, FPAR_NPV_ increased with increase of PAI. However, when PAI was greater than 2.0, negative relations between FPAR_NPV_ and PAI can be found. For a given PAI, green FPAR increased, while FPAR_NPV_ and the total FPAR decreased along with the increase of the proportion of leaf area in the canopy. The maximum and minimum values of green FPAR were 0.9562, 0.0323 respectively, and those of the total FPAR were 0.9674, 0.0647 respectively. The minimum value of FPAR_NPV_ occurred when PAI was smallest (0.1) and the proportion of leaf area was highest (98%). The maximum of FPAR_NPV_ was 0.3257, which occurred at PAI of 2.0 and the proportion of leaf area of 50%.

The total FPAR, FPAR_PAV_, and FPAR_NPV_ of evergreen coniferous forests had similar modes with those of broadleaf deciduous forests (Figure 2). The total FPAR and green FPAR increased as canopy plant area index was increasing. At a plant area index of 0.1, their minimum values occurred, and they had maximum values when PAI was 7.0. Canopy FPAR_NPV_ had more complicate trajectories, and at a PAI of 2.0, the maximum values of FPAR_NPV_ occurred.

For evergreen coniferous forests and broadleaf deciduous forests, the only difference in FPAR was the size of their values. The maximum values of FPAR_PAV_, FPAR_NPV_, and total FPAR for evergreen coniferous forests were 0.9515 (PAI, 7.0, proportion of leaf area, 98%), 0.323 (PAI, 2.0, proportion of leaf area, 50%), 0.963 (PAI, 7.0, proportion of leaf area, 50%), respectively. And their minimum values were 0.0316 (PAI, 0.1, proportion of leaf area, 50%), 0.0016 (PAI, 0.1, proportion of leaf area, 98%), 0.0631 (PAI, 0.1, proportion of leaf area, 98%).

Four vegetation indices (NDVI, EVI, EVI2, and SAVI) of deciduous broadleaf forests were shown in Figure 3. As PAI was 7.0 and the proportion of leaf area was 98%, NDVI, EVI, EVI2, and SAVI had their maximum values, which were 0.89, 0.69, 0.66, and 0.62, respectively. It is similar with the total FAPR and green FPAR. Of four vegetation indices, NDVI was generally greater than other three vegetation indices, and SAVI value was the least for a given canopy structure. Compared with EVI, EVI2, and SAVI, NDVI displayed less sensibility to the variability of proportion of leaf area in a canopy. Maximum increment of NDVI was only 0.035 when the proportion of leaf area increased from 50% to 98%, while for EVI, EVI2, SAVI, and the maximum increments were 0.084, 0.086, 0.069, respectively. The sizes of increments were related to canopy PAI. When PAI was 3.0, the influence of variation in the proportion of leaf area on vegetation indices was most significant. Variability of vegetation indices decreased with increase of the proportion of leaf area in a canopy as PAI was less or greater than 3.0. Compared with the proportion of leaf area, the canopy PAI had more significant influence on four vegetation indices. For instance, at the proportion of leaf area of 98%, NDVI, EVI, EVI2, and SAVI increased by 0.71, 0.57, 0.51, 0.47 respectively with PAI increasing from 0.1 to 7.0. The increments decreased when the proportion of leaf area decreased. The increments in NDVI, EVI, EVI2, and SAVI were 0.69, 0.52, 0.47, and 0.43 respectively at the proportion of leaf area of 50%, which were smaller than those at the proportion of leaf area of 98%.For evergreen coniferous forests, their vegetation indices had similar dynamics with those of broadleaf deciduous forests (Figure 4). The only differences were values of vegetation indices. Four vegetation indices had their maximum when PAI and the proportion of leaf area in canopies were the highest.

### 2.2. Relationship between FPAR and Vegetation Indices

Figure 5 demonstrates the variations of canopy FPAR (including FPAR_PAV_, FPAR_NPV_ and the total FPAR) with four vegetation indices for broadleaf deciduous forests. The total FPAR, FPAR_PAV_ and FPAR_NPV_ had different change patterns with vegetation indices. The total FPAR and FPAR_PAV_ increased with vegetation indices regardless of the proportion of leaf area. For FPAR_NPV_, its variations with vegetation indices were more complex than the total FPAR and FPAR_PAV_. With the increase of vegetation indices, FPAR_NPV_ increased initially (PAI < 0.2) and thereafter decreased (PAI > 0.2). Furthermore, the trajectory of FPAR-vegetation index varied with the proportion of leaf area in 2-D space composed of FPAR and vegetation index. For a given value of anyone of vegetation indices, FPAR_PAV_ values increased with the proportion of leaf area in canopies, while the total FPAR and FPAR_NPV_ decreased. Pearson correlation coefficients between the total FPAR, FPAR_PAV_ and each vegetation index were calculated, and they were high and can pass the test at the 0.01 significance level according to Pearson correlation critical values table. This indicated there were significant linear positive correlations between them. However, correlation coefficients between FPAR_NPV_ and four vegetation indices were low (r < 0.12). Particularly, the correlation coefficient between FPAR_NPV_ and EVI2 was only 0.019. NDVI had closer relationship with the total FAPR than other three vegetation indices. EVI demonstrated better consistence with FPAR_PAV_. When the proportions of leaf area rose, the correlation coefficients between FPAR and four vegetation indices increased. For example, the correlation coefficient between FPAR_PAV_ and NDVI, EVI, EVI2, SAVI decreased by 0.03, 0.015, 0.017, 0.022 when the proportions of leaf area in canopies varied from 0.5 to 0.98.

For evergreen coniferous forests, the relationships between the canopy FPAR (including FPAR_PAV_, FPAR_NPV_ and total FPAR) and four vegetation indices were similar with those of broadleaf deciduous forests (Figure 6). The significant linear relationship can be found between canopy FPAR (including the total FPAR, FPAR_PAV_) and each vegetation index, while the relationship between FPAR_NPV_ and each vegetation index was nonlinear. For evergreen coniferous, the total FPAR and FPAR_PAV_ had the closest relation with NDVI (r = 0.993), EVI (r = 0.99), respectively, but correlation coefficients FPAR_NPV_ between and four vegetation indices were less than 0.07. With the increase of the proportion of leaf are in canopies, correlation coefficients between FPAR_PAV_ and four vegetation indices increased.

### 2.3. Influence Factors of Green FPAR and Vegetation Indices

Green FPAR (FPAR_PAV_) is smaller than the total FPAR which includes light absorption of all components in the canopy. As well as green leaves, non-green components also absorb solar photons. Unfortunately, the energy is not used for photosynthesis. Under the given condition in this study, the ratio of FPAR_NPV_ to the total FPAR varied from 0.26 percent to 56.93 percent in deciduous broadleaf forests, while the ratio varied from 0.26 percent to 56.89 percent in evergreen coniferous forests. The contributions from non-green components to the total FPAR are similar for two different forests ecosystems, but it varies with canopy structure. In order to more clearly illustrate the variations of FPAR_NPV_ with canopy structure, we simulated green FPAR of six tree species based on spectrum of theirs components (Figure 7). It can be found that non-green contributions increased along with the increase of proportion of non-photosynthetic components in the canopy and the decrease of forests PAI. FPAR_NPV_ values of six tree species were no more than 0.003 at PAI of 7 and the proportion of green component of 0.02. Even though non-green components accounted for a higher proportion in the canopy, FPAR_NPV_ was still low if PAI is high. For instance, at a PAI of 7, FPAR_NPV_ increased only by 0.12 when the proportion of non-green components varied from 2% to 50%. However, when PAI was below 1.0 and the proportion of non-green components in the canopy was 0.5, non-green components had more contributions than green leaves. This suggests that non-green contributions are more significant in forests with low PAI values and a large proportion of non-green components in canopies, while in forests with high vegetation coverage green contributions to the total FPAR will higher. Mature forests, whether broadleaf forests ecosystems or coniferous forests ecosystems, have generally closed canopies and high PAI values during the peak growing season. And at the time, the canopies are mainly dominated by leaves, and there are only a very small proportion of non-green components. In addition, leaves are located on the top of the canopy, so solar photons will firstly interact with leaves once light arrives at the canopy. In PAR spectral regions (400 nm–700 nm), solar photons in the visible regions are strongly absorbed by chlorophyll and carotenoids of green leaves [30]. Only photons scattered (reflected and transmitted) by green leaves have chance to interact with non-green woody components. If there is a dense leaf layer, leaves will intercept most of photons, and thus only little PAR can be absorbed by woody material. This may explain why FPAR_PAV_ has high values, while FPAR_NPV_ is so low in forests ecosystems with high PAI values and small proportion of nonphotosynthetic components. However, for open canopies, there are more gaps in canopies and solar photons can arrive at non-green components directly and interact with branches. In addition to direct sunlight, non-green components absorb light transmitted by leaves and reflected by ground. Non-green components absorb solar light without any transmission [31]. These factors make non-green components absorb more sunlight than green leaves in some cases.

Canopies of six tree species with identical canopy structure, background and illumination conditions had different FPAR_PAV_. Oak had the highest green FPAR values, and Alder followed it. PAR by absorbed by green leaves of Juniper was the least. When the canopy structure, background and illumination conditions were same, the only difference between tree species was the absorption capacity of green leaves. According to spectral data measured in the laboratory, Oak had the largest absorptance in PAR regions, and the second was Alder (Figure 8). There is a positive relationship between leaf absorptance and green FPAR.

Thus, canopy PAI, the proportion and optical properties of canopy components have impacts on FPAR_PAV_, but impacts of these factors are different. To compare impacts of three factors, we selected FPAR_PAV_ values in different scenarios including three PAI values, two proportions and six tree species (Table 1). The maximum difference in FPAR_PAV_ between six tree species was 0.0228 (Oak and Juniper) which occurred when the proportion of green leaves was 98% and PAI is 1. The maximum difference (Oak and Juniper) was only 0.0199 when the proportion of green leaves was 50%. However, when PAI increased from 1 to 4, green FPAR increased by at least 0.4039 (Juniper). It seems that leaf absorptance has less impact on green FPAR than PAI. The possible reason is that there is small difference in the absorptance of green leaves between tree species. When the proportions of green leaves in the canopy varied from 50% to 98%, FPAR_PAV_ increased by more than 0.1 (PAI was 7), and the increments were large as PAI was 4, more than 0.2. The impacts of the proportion of canopy components were less than those of PAI, but more than leaf absorptance in the PAR region.

For four our vegetation indices, they varied with canopy structure. They increased with rising plant area index and the proportion of green component in the canopy. The theoretical basis for these vegetation indices lies with the red-NIR contrast of vegetation spectral reflectance signatures. In the red region, live vegetation has a strong absorption. In the NIR regions, leaves exhibit large scattering (reflecting and transmitting) but branches have no transmission. Consequently, the absorption of solar light in the red region will increase and the reflectance will decrease when plant area index rises. However, the light scattered in the NIR region also increase and the reflectance increase with the increase of plant area index. This makes four vegetation indices increase. If there are more green leaves, the reflectance in the red region rises but in the NIR region has opposite result. Non-green components can increase the magnitude of NIR absorbance because of its negligible NIR transmittance [31].

### 2.4. Estimation of Green FPAR

Presently, FPAR is mainly estimated based on empirical models (e.g., spectral vegetation indices, LAI) and radiative transfer model. Spectral vegetation indices play a significantly important role in FPAR inversion [32]. Similarly with the total FPAR, green FPAR also demonstrates strong linear correlations with four vegetation indices (NDVI, EVI, EVI2, and SAVI). The difference in correlation coefficients between green FPAR and four vegetation indices is small. But compared with other three vegetation indices, EVI has higher correlation coefficients with green FPAR. In an earlier study, EVI is founded that it is linearly correlated with the green leaf area index (LAI) in crop fields based on airborne multispectral data [33]. Additionally, the seasonal dynamics of EVI better mimic those of GPP in terms of phase and amplitude [32]. Therefore, if spectral vegetation indices are used to estimate the PAR absorbed by green leaves, EVI should be a good choice. Nevertheless, there is a problem solved, namely the canopy structure variations. Given an invariant proportion of green components, the empirical model between vegetation indices and FPAR can be constructed easily. Unfortunately, the ratio of green to non-green components in the deciduous broadleaf forest canopy is seasonal. And only in forests with a high vegetation index, the impacts of the variation of the proportion of canopy components on relationships between green FPAR and vegetation indices is negligible. It seems that to accurately estimate green FPAR in forests, the ratio of components must be known. However, it is a very difficult task to quantify green and non-green components at the canopy levels. For a mature deciduous forest, PAI values increase rapidly with leaf expansion and non-green components growth after the spring growing season begins. Compared with green leaves, the variations of non-green components are small, and the increase of PAI mainly comes from leaf development [34]. Therefore, in a year, non-green components can be regarded as a constant. In winter, there are only non-green components in the canopy, so the PAI in this period can represent NPV area index. According to some studies, PAI of deciduous broadleaf forests in winter is about 0.5 [35,36]. While, mature evergreen coniferous forests are different from deciduous broadleaf forests. Evergreen forests can preserve a high PAI and a low proportion of non-green components all the year round. Consequently, the proportion of canopy components can be regarded a constant. Field campaigns found that the proportion of needle leaves in the canopy is generally 90% [23].

According to above analysis, we re-simulated the canopy FPAR and reflectance using SAIL model. In the new simulation, NPV area index of deciduous broadleaf forests and the proportion of green components of evergreen coniferous forests were set as 0.5, 90%, respectively and PAI varied from 1 to 7, while other parameters were preserved. The results are shown in Figure 9. The Among four vegetation indices, EVI shows the highest correlations with FPAR_PAV_ (r = 0.99), while NDVI has the lowest correlations with green FPAR. It is well known that EVI, EVI2, and SAVI are modification or optimization versions of NDVI [16,37,38]. The fundamental objective of optimization is to reduce atmospheric or soil effects and to isolate the green photosynthetically-active signal from the mixed’ and ‘contaminated pixels [39]. Consequently, the three vegetation indices show better relationships with green FPAR than NDVI.

Through the linear regression method, the formulas between ${FPAR}_{PAV}$ and EVI for deciduous broadleaf forests and evergreen coniferous forests were constructed. Their ${FPAR}_{PAV}$ can be represented as respectively
$${FPAR}_{PAV,DBF}=2.31\times EVI-0.67$$
$${FPAR}_{PAV,ENF}=2.37\times EVI-0.34$$
where, DBF is deciduous broadleaf forests, ENF is evergreen coniferous forests.

The errors were small when ${FPAR}_{PAV}$ was estimated through the model. For deciduous broadleaf forests, the mean error of ${FPAR}_{PAV}$ was 0.0086, and the error rate was 1.99%, while the mean error was 0.026 and the error rate was 3.76% for evergreen coniferous forests. It seems that the inversion model for deciduous broadleaf forests can produce better results than the model for evergreen coniferous forests.

## 3. Materials and Methods

### 3.1. SAIL Model

The SAIL model is a canopy radiative transfer model developed by Verhoef through extending the Suits model to allow for the variations of leaf angles. The model assumes that the canopy is consist of small absorbing and scattering elements, with known optical properties, distributed randomly in horizontal layers and with a known angular distribution [40]. It permits to calculate the bidirectional reflectance factor and FPAR of plant canopies by solving the scattering and absorption of four upward/downward radiative fluxes [41,42]. The four-stream differential equations form the basis of the SAIL models. These equations describe the contribution of the vegetation components illuminated by three fluxes ($E_{s},E_{-},E_{+}$) to the flux toward the observer ($E_{0}$) and the attenuation of this flux along the path:$$dE_{s}/dx={kE}_{s}$$
$$dE_{-}/dx={-sE}_{s}+aE_{-}-\sigma E_{+}$$
$$dE_{+}/dx={s^{\prime }E}_{s}+\sigma E_{-}-aE_{+}$$
$$dE_{0}/dx={wE}_{s}+vE_{-}+uE_{+}-KE_{0}$$
where, $E_{s}$ is direct solar irradiance, E_− is diffuse downward irradiance, $E_{+}$ is diffuse upward irradiance, $E_{0}$ is radiance in the direction of view. The detailed calculated can be found in reference [22].

The SAIL model requires only a few parameters. Evaluations of the SAIL model have shown generally adequate agreement with observations. Presently, the SAIL model has been widely used in remote sensing research for investigating and describing absorption and reflectance properties of vegetation canopies [43,44]. In the SAIL model, all components in the canopy are randomly mixed in a single vegetation layer. As a result, the output FPAR is the canopy total FPAR, and rather than the FPAR of each component. As well known, in the SAIL model the canopy is assumed as a number of layers which are made up of small flat leaves in the SAIL model. The configuration provides a means to separate the FPAR for different components. If a canopy is divided into multiple layers and each layer only contains one component during simulation, the FPAR in each layer can be calculated, respectively. Thus, the radiation absorbed by green leaves can be separated from the total FPAR. Based on this point, the SAIL model was reconfigured in our study. In the revised model, FPAR for each layer can be calculated from the three flux streams as the balance between energy into and out of that layer. For example, FPAR the first layer can be described as following
$$FPAR=FLUX3\left(1\right)+FLUX2\left(1\right)-FLUX1\left(1\right)-FLUX3\left(2\right)-FLUX2\left(2\right)+FLUX1(2)$$ where, FLUX1 is the upward diffuse flux; FLUX2 is the downward diffuse flux; FLUX3 is the downward specular flux; The number in bracket denotes the layer number (e.g., 1 is the top layer, 2 is the second layer…).

### 3.2. Model Parameterization

The input parameters of the revised SAIL model include the number of canopy layers, reflectance, transmittance and leaf inclination angle distribution of canopy components, background reflectance, leaf area index of each component, the fraction of direct irradiance, solar zenith angle, view zenith angle and view azimuth angle (Table 2). Leaf area index were calculated according to the plant area index (include leaf and branch) and the proportion of leaf. In our studies, six proportion values of green component (0.5, 0.6, 0.7, 0.8, 0.9, 0.95, and 0.98) were set.

In this study, broadleaf and coniferous forests were considered. The forest spectral data comes from the Superior National Forest study. Six tree species including red alder, white oak, aspen, western hemlock, western juniper and sitka spruce were selected. The leaf and branch spectra of broadleaf tree species and needle leaf tree species were averaged respectively (Figure 10). These averaged spectra inputted into the SAIL model to simulate reflectance and FPAR for broadleaf and needle leaf canopies.

### 3.3. Vegetation Index

Using spectral vegetation indices to estimate the canopy FPAR is a frequently-used approach in the remote sensing community [45,46,47,48]. NDVI, derived from the red and near-infrared reflectance [49], is the most-used vegetation index for estimation of the canopy FPAR [13,16]. However, it is well known that NDVI has several limitations, including saturation in a closed canopy and sensitivity to both atmospheric aerosols and the soil background [39,50]. Therefore, a number of derivatives and alternatives to NDVI have been proposed in the scientific literature to address these limitations, including the soil-adjusted vegetation index (SAVI), the enhanced vegetation index (EVI), EVI2 and so on. SAVI minimizes soil brightness influences from spectral vegetation indices [37]. EVI is an optimized vegetation index, and it enhances the vegetation signal with improved sensitivity in high biomass regions and vegetation monitoring through a de-coupling of the canopy background signal and a reduction in atmosphere influences [38]. Different from NDVI and SAVI, EVI is calculated based on three bands. A blue band is needed in addition to the red and near-infrared bands, which make EVI difficult to generate long-term EVI time series as NDVI counterpart. Therefore, EVI2 without a blue band, which has the best similarity with the 3-band EVI, is developed by Jiang et al. [51]. These vegetation indices are calculated according to following equations respectively.
$$NDVI=\frac{R_{NIR}-R_{RED}}{R_{NIR}+R_{RED}}$$
$$EVI=2.5\times \frac{R_{NIR}-R_{RED}}{1+R_{NIR}+6\times R_{RED}-7.5\times R_{BLUE}}$$
$$EVI2=\frac{R_{NIR}-R_{RED}}{R_{NIR}+2.4\times R_{RED}+1}$$
$$SAVI=(1+L)\times \frac{R_{NIR}-R_{RED}}{R_{NIR}+R_{RED}+L}$$
where $R_{NIR}$, $R_{RED}$, $R_{BLUE}$ denote the spectral reflectance measurements acquired in the red, near-infrared, blue regions, respectively; L is a canopy background adjustment factor and was set as 0.5.

Potentially, these vegetation indices vary from −1 to 1.

## 4. Conclusions

Forest canopies are composed of green and non-green components with different optical properties and functions. Green FPAR determined by green component in the canopy represents canopy photosynthetic capacity, so it is more important than the total FPAR in carbon cycle research. In this study, we used a modified version of SAIL model to study light absorption by forests canopies (including evergreen coniferous forests and deciduous broadleaf forests). The new SAIL model can distinguish the PAR absorbed by the canopy green and non-green components. The canopy FPAR including the total FPAR, green and non-green FPAR (FPAR_NPV_) were simulated in the given scenarios.

Green FPAR varies with canopy structure. Plant area index, the proportion of canopy components, optical properties of components have impacts on green FPAR. Plant area index is the most important factor influencing the magnitude of green FPAR among three variables. The proportion of canopy components is second. Green FPAR rises with the increase of plant area index and the proportion of green leaves. In the dense forests, green FPAR is close to the total FPAR, but in open canopies, non-green components have higher contribution to the total FPAR. There are significant relationships between green FPAR and four vegetation indices. Compared with NDVI, EVI2 and SAVI, EVI has a higher correlation coefficient with green FPAR. It may be a good indicator of green FPAR.

There is currently a lack of in situ independent data for evaluating green FPAR, due to the difficulty of measurement in the field. This study represents our effort in using a radiative transfer model to partition the canopy FPAR into green FPAR and FPAR_NPV_, and provides a method to estimate green FPAR. For two typical forests, two formulas to calculate green FPAR are constructed, respectively. It should be noted that the results of the model have not been validated due to the lack of field green FPAR. In this study, only three variables (plant area index, the optical properties and proportion of canopy components) are discussed in the simulation. It is not enough because factors influencing canopy absorption and optical properties are various. More variables need to be considered when simulating in future study. Moreover, forest background and air conditions are not considered and discussed in the study. In open canopies, there are more photons through canopy gap on the soil and these photons are reflected. Therefore, the effect of background may be significant. Solar photons reflected by canopies are attenuated by clouds, aerosols and molecules along its way to satellite sensors. Though remotely sensed data can be corrected through atmospheric correction algorithms, some residual effects still exist. Moreover, it is also difficult to obtain accurate atmospheric correction parameters. Consequently, to invert green FPAR using remotely sensed data using vegetation indices, atmospherically resistant vegetation indices is needed. These factors will be discussed in future studies.

## Figures and Tables

**Figure 1 plants-12-01927-f001:**
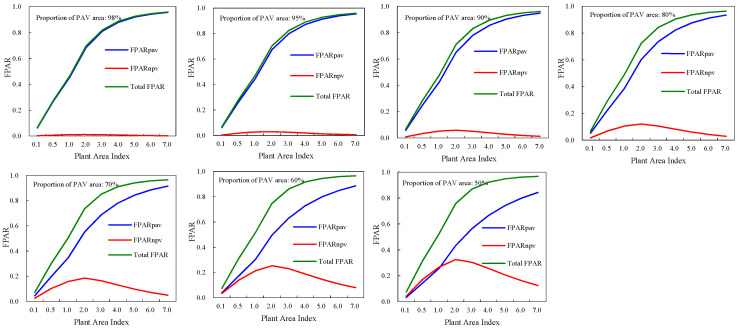
FPAR of broadleaf deciduous forests canopies from the SAIL model.

**Figure 2 plants-12-01927-f002:**
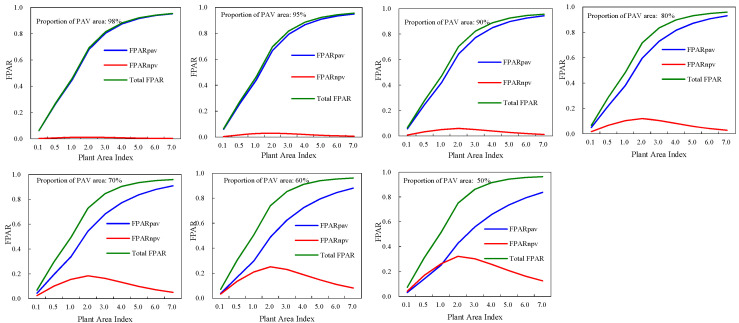
FPAR of evergreen coniferous forests canopies from the SAIL model.

**Figure 3 plants-12-01927-f003:**
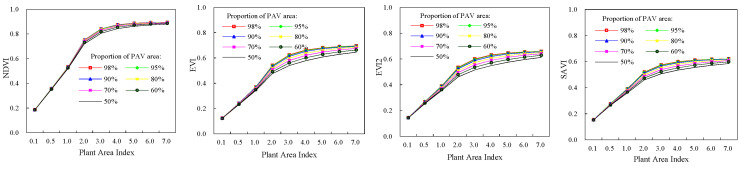
Vegetation indices of broadleaf deciduous forests.

**Figure 4 plants-12-01927-f004:**
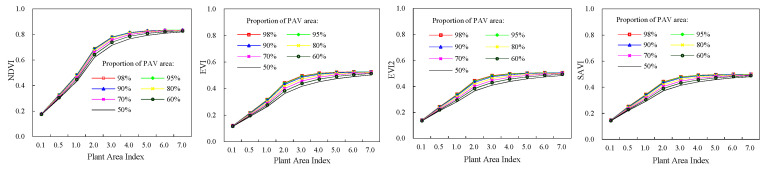
Vegetation indices of evergreen coniferous forests.

**Figure 5 plants-12-01927-f005:**
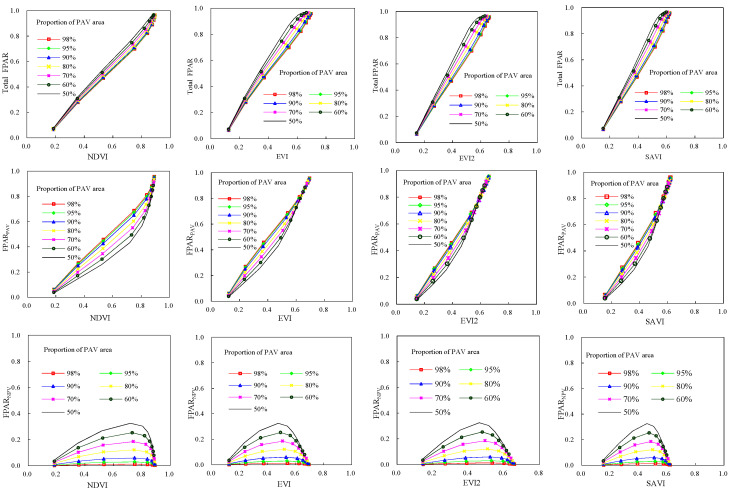
The relationship between FPAR and vegetation indices for broadleaf deciduous forests.

**Figure 6 plants-12-01927-f006:**
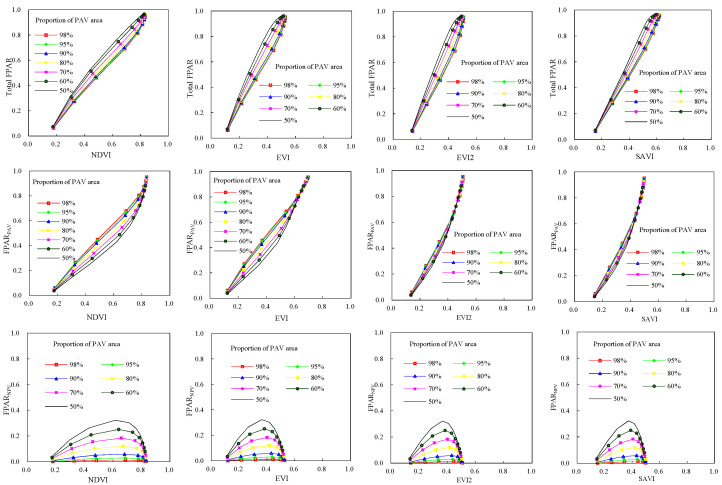
The relationship between FPAR and vegetation indices for evergreen coniferous forest.

**Figure 7 plants-12-01927-f007:**
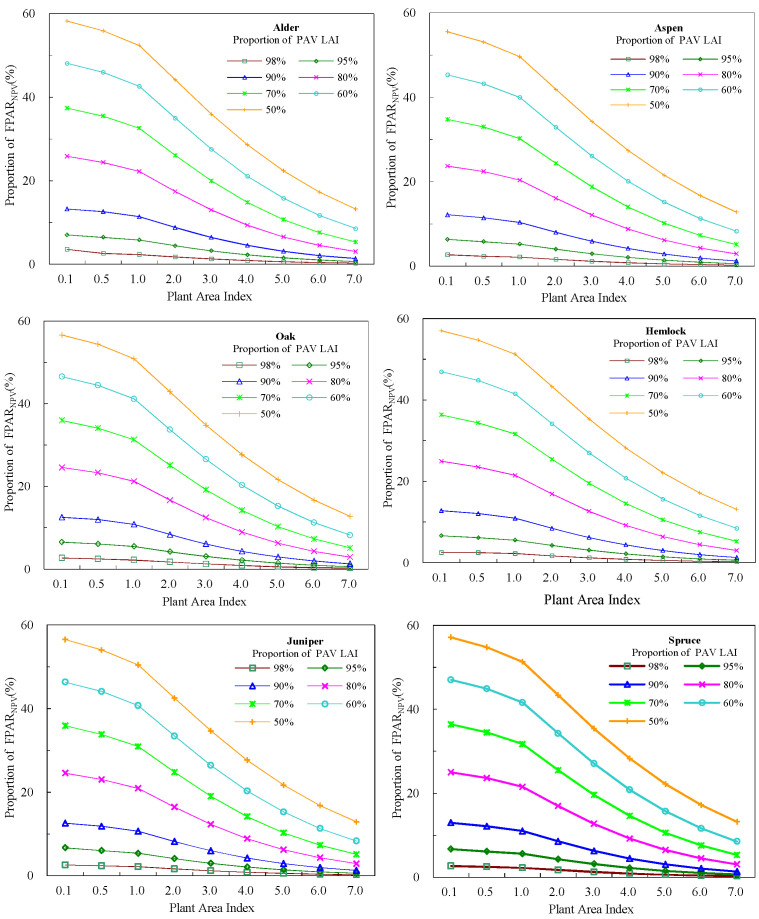
Variations of green FPAR with plan area index for six tree species.

**Figure 8 plants-12-01927-f008:**
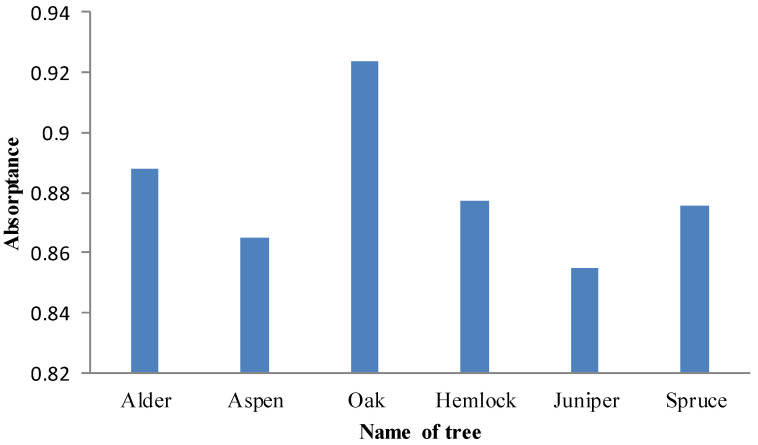
Absorptance of different tree species.

**Figure 9 plants-12-01927-f009:**
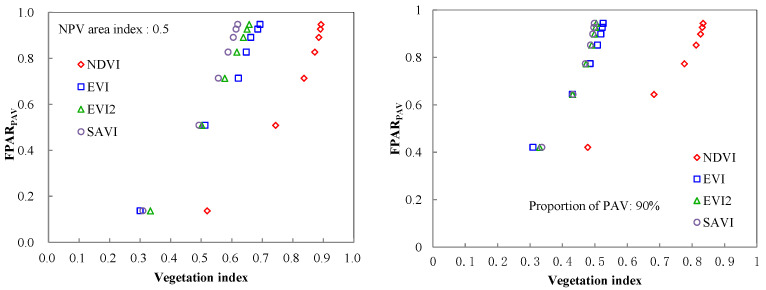
The relations between green FPAR and spectral vegetation index ((**left**), deciduous. broadleaf forests; (**right**), evergreen coniferous forests).

**Figure 10 plants-12-01927-f010:**
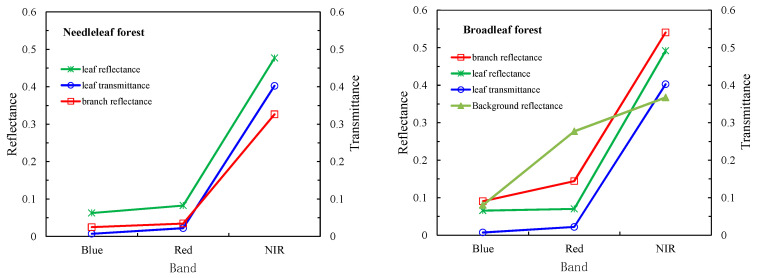
The spectrum of canopy components of needle leaf, broadleaf forests and background.

**Table 1 plants-12-01927-t001:** Green FPAR of different species in give scenarios.

Tree Species		98%			50%		
	PAI	1	4	7	1	4	7
Alder		0.4536	0.8798	0.9566	0.2541	0.6619	0.8409
Aspen		0.4495	0.8754	0.9523	0.2558	0.6645	0.8398
Oak		0.4678	0.8891	0.9628	0.2622	0.6725	0.8497
Hemlock		0.456	0.878	0.9533	0.2525	0.6577	0.8374
Juniper		0.445	0.8669	0.9415	0.2516	0.6555	0.8298
Spruce		0.4531	0.8797	0.9568	0.2537	0.6614	0.8406

**Table 2 plants-12-01927-t002:** Input parameters in SAIL model.

Parameter	Value
Canopy architecture	
Inclination angle distribution	Spherical (leaf) Planophile (branch)
Plant area index	0.1, 0.5, 1, 2, 3, 4, 5, 6, 7
Illumination and viewing geometry	
Solar zenith angle/(°)	30
Fraction of direct irradiance	1.0
View azimuth angle/(°)	0
View zenith angle/(°)	0

## Data Availability

The data used to support the findings of this study are available from the corresponding author upon request.

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
