# Peer review of "An Assessment of Relations between Vegetation Green FPAR and Vegetation Indices through a Radiative Transfer Model"

_plants, 2023, doi:10.3390/plants12101927_

Round 1
Reviewer 1 Report
The manuscript is written in understandable language. The task of the manuscriot is relevant and the results are well presented. Some doubts concern the list of references. Unfortunately, in the Introduction, the authors use references to rather "old" scientific studies (not the last 5 years). This is not entirely clear. In general, the analysis of the list of sources, presented by the authors, made it possible to establish, that 40% of this list are works of the last century. Another 40% are studies 10 or more years old. We invite the authors to work with the literature and present the latest achievements in this area.
In general, the manuscript leaves a good impression and can be recommended after a minor revision.
Author Response
Thank you sincerely for your advice. You advice improves the quality of the study, and furthermore, provides some ideas for further studying. Thank you again!!
Thank you sincerely for your advice. You advice improves the quality of the study, and furthermore, provides some ideas for further studying. Thank you again!!
In the new manuscript, we rewrote the introduction, and some new references were adopted.
“Photosynthetically active radiation (PAR) refers to the radiation with the spectral range from 400-700nm that is used by plants in photosynthesis. The fraction of absorbed photosynthetically active radiation (FPAR) is the ratio of the PAR absorbed by vegetation to the PAR across an integrated plant canopy [1]. FPAR is closely linked to canopy functioning processes such as canopy photosynthesis, carbon assimilation and evapotranspiration rates. It is a key biophysical variable for vegetation productivity estimation, vegetation growth condition monitoring, and climate change analysis [2]. An accurate specification of FPAR is an important detail in large scale productivity and carbon budget models [3]. FPAR is recognized by the global climate observing system (GCOS) as essential climate variables (ECVs) [4].
Traditionally, the canopy total FPAR is directly measured in the field through optical instruments including hemispherical photographs [5], the tracking radiation and architecture of canopies (TRAC) instrument [6], the burr Brown data acquisition system (BBDAS) [7].Traditional method is consuming and expensive. This makes assessment of FAPR on the landscape and ecosystem scales impractical. As an alternative solution, FPAR can be derived through remote measures of surface spectral reflectance, and remote sensing-based methods are currently the only feasible way of acquiring FPAR estimates at the temporal and spatial scales necessary for biogeochemical, biosphere-atmosphere and climate models [8-11].A number of state of the art algorithms have been proposed to estimate the important environmental variable. They can be categorized as , (a) empirical methods based on relationships between FPAR and vegetation indices, LAI(Leaf area index) ;(b) physical methods based on the physics of radiation interaction with elements of a canopy and transport within the vegetative medium;(c) machine learning algorithms[12,13]. Moreover, the number of researches utilizing hybrid regression methods combining radiative transfer model simulations with machine learning regression methods to derive vegetation FPAR is increasing [14]. During the recent decades, a series of satellite FPAR products have been developed based on different definitions, assumptions, retrieval algorithms, and sensors including MODerate-resolution Imaging Spectroradiometer (MODIS), Multi-angle Imaging SpectroRadiometer(MISR), MEdium Resolution Imaging Spectrometer(MERIS),Sea-Viewing Wide Field-of-View Sensor (SeaWiFS),GEOV1,Global Land Surface Satellite (GLASS),CYCLOPES,Visible Infrared Imaging Radiometer (VIIRS), FPAR3g, Copernicus Global Land Service (CGLS) ,Earth Polychromatic Imaging Camera (EPIC) and Sentinel-3 Ocean and Land Colour Instrument (OLCI) [13,15].
For woody plants, the canopy is composed of photosynthetically active vegetation (PAV, mostly green leaves) and non-photosynthetically active vegetation (NPV, e.g., braches, stems) [16]. PAV and NPV absorb PAR, but only the PAR absorbed by PAV is used for photosynthesis [17, 18]. Obviously, this quantity of FPAR determined by PAV is lower than the canopy total FPAR because it does not include PAR absorption by wood materials. For forest ecosystems, NPV can increase canopy FPAR by 10–40% when leaf area index is less than 3.0 [8]. If the NPV contribution to canopy FPAR is not removed, the PAR used to photosynthesis will be overestimated [19]. Additionally, the total FPAR cannot accurately reflect the spatiotemporal variations in photosynthesis due to the varying fractions of NPV [20,21]. Therefore, for the canopy consist of PAV and NPV, the FPAR should be partitioned into FPARPAV(green FPAR) and FPARNPV, and FPARPAV should be estimated instead of the canopy total FPAR in order to improve the estimation accuracy of vegetation productivity. Any model that accounts for FAPARPAV is likely to substantially improve estimation of GPP or NPP of forests, given a known value of light-use efficiency [22].However, there is no practical direct method of separating the PAR absorbed by the green leaves [23,24]. In some researches, green FPAR is defined as the product of the total FPAR and the fraction of leaf area in the canopy [2,18]. That approach is based on the idea that the green leaves and NPV have the same spectral characteristics. To forest canopies, NPV is significantly different from green leaves, which makes the approach invalid. Fortunately, green FPAR can be calculated from radiative transfer models describing the transfer of solar radiation in plant canopies [25,26]. Physics-based radiative transfer models represent the scattering and absorption of radiation by scattering elements of canopies. These models are helpful for scientists to understand how radiations interact with the environment, and how they propagate towards the sensor. Radiative transfer models play key roles in FPAR inversion based on remotely sensed data. Such an approach calculating through radiative transfer models is time consuming [20]. Vegetation indices play key roles in FPAR inversion based on remotely sensed data. Live green plants strongly absorb solar radiation in red spectral region, and scatter (reflect and transmit) solar radiation in the near-infrared spectral region. Vegetation indices derived from radiometric measurements in red and near-infrared wavelengths can reflect absorption and reflection of vegetation to solar radiation. Consequently, some vegetation indices (e.g. NDVI,EVI )are related to FPAR, and are frequently used to estimated FPAR due to its ease of use and simplicity [13,26-29]. Relationships between vegetation indices and FPARPAV need be assessed for forest ecosystems.
In this study, the scattering by arbitrary inclined leaves model (SAIL model) was used to partition the canopy FPAR. The aims of our study are twofold, (I) to simulate and analyze the variation of green FPAR with the canopy structure; (II) to explore the relations between the green FPAR and vegetation indices for estimation of green FPAR .”
The references added are as follows:
Dong, T.; Wu, B.; Meng, J.; Xin, D.; Shang, J. Sensitivity analysis of retrieving fraction of absorbed photosynthetically active radiation (fpar) using remote sensing data. Acta Ecologica Sinica 2016,36; 1-7.
Verrelst, J. Z.; Malenovský, C.; Van der Tol, G.; Camps-Valls, J. P.; Gastellu-Etchegorry, P.; Lewis, P.; North,P.; Moreno,J.. Quantifying vegetation biophysical variables from imaging spectroscopy data: A review on retrieval methods. Surveys in Geophysics, 2018, 40: 589-629.
Zhang, Z.; Zhang, Y. ;Zhang, Y. ;Gobron, N.; Li, Z. The potential of satellite fpar product for gpp estimation: an indirect evaluation using solar-induced chlorophyll fluorescence. Remote Sens. Environ. 2020, 240: 111686.
Zhang, Q.; Cheng, Y. B.; Lyapustin, A.I.; Wang, Y.;Gao, F.; Suyker, A.; Verma,S.; Middleton, E.M.; Estimation of crop gross primary production (GPP): fAPARchl versus MOD15A2 FPAR. Remote Sens. Environ 2014,153, 1–6.
Zhang, Y.; Xiao, X.; Wolf, S.; Wu, J.; Wu, X.; Gioli, B.; Wohlfahrt, G.; Cescatti,A. van der Tol, C.; Zhou, S.; Gough, C.M.; Gentine, P.; Zhang, Y.G.; Steinbrecher, R.; Ardö, J. Spatio-temporal convergence of maximum daily light-use efficiency based on radiation absorption by canopy chlorophyll. Geophys. Res. Lett 2018, 45, 3508–3519.
Li, L.; Xin, X.Z.; Tang, Y.; Bai, J.H.; Du, Y.M.; Sun, L.; Wen, J.G; Zhong, B.; Wu, S.L.; Zhang, H.L.; Yu, S.S.; Liu, Q.H. Fraction of absorbed photosynthetically active radiation inversion algorithm of GF-1 data combining radiative transfer model simulation and deep learning. National Remote Sensing Bulletin 2023, 27:700-710
Tian, D.F.; Fan, W.J.; Ren, H.Z. Progress of fraction of absorbed photosynthetically active radiation retrieval from remote sensing data. J. Remote Sensi.2020, 24:1307-1324
Ye,Y.Y.; Qi, J.B.; Cao,Y.; Jiang, J.Y. Relationship between FPARgreen and several vegetation indices in heterogeneous vegetation based on LESS model. Remote Sensing Technology and Application 2023, 38:51-65.
Rahman, M.M.; Lamb, D.W.; Stanley, J.N. The impact of solar illumination angle when using active optical sensing of NDVI to infer FAPAR in a pasture canopy. Agr. Forest Meteorol. 2015, 202: 39-43
Yuan, Y.R.;Wang,J.Y.;Yang,J.W.;Xiong, J.N. Research on FPAR Estimation of Wetland in Zoige Plateau based on Vegetation Index. Remote Sensing Technology and Application 2022, 37: 1267-1276
Pinty, B.; Lavergne, T.; Widlowski,J.L.;Gobron,N.; Verstraete, M. M. On the need to observe vegetation canopies in the near-infrared to estimate visible light absorption. Remote Sens. Environ. 2009,113: 10-23
Dong, H.; He, F.J.; Zhang, C.F. Relationship between FPAR green and several vegetation indices based on radiative transfer model. Journal of Huazhong Agricultural University 2016,35: 70-75.
Zhao, Y.H.; Hou, P.; Jiang, J.B.; Jiang,Y.; Zhang, B.;Bai, J.J.; Xu, H.T. Progress in quantitative inversion of vegetation ecological remote sensing parameters. National Remote Sensing Bulletin 2021,25:2173-2197

Reviewer 2 Report
Most of the work presents the results of simulations performed with a model (SAIL model) that already exists in the literature and, therefore, without adding any element of originality.
The only original elaboration is described in section 4.2, but it is poorly presented, without statistical analysis.
The text is not always comprehensive, and the organization of the whole paper is confusing. For example:
- The description of the characteristics of the SAIL model in 2.1 is deficient and formula (1) is not clear enough.
- Figure 1 is not referred to or commented on in the text. Also, it is not clear why two figures are needed and why “branch reflectance” is repeated twice.
- All the figures are barely legible.
- Figure 2 has a different caption than the others “Percent PAV LAI” vs “Proportion of PAV area”. Furthermore, there is an extra image (Plant Area Index vs Total FPAR) which is not present in figure 3 and is not discussed. Why is total FPAR not sensitive to LAI? Check the spelling of the X-axis label (Plant not Pland).
- In 3.2 you write about significant correlations, but no statistical analysis is reported.
- Figure 6. Axes ranges should be the same.
- In the discussion more results of simulations are reported.
- Again, at the beginning of 4.2, you write about correlations without any statistical analysis. Furthermore, no statistical analysis is reported for your interpolated formulas.
I suggest review by a professional native speaker
Author Response
Thank you sincerely for your advice. You advice improves the quality of the study. We revised the manuscript according to your advice. Thank you again!!
The figure reprocessed can be found in the word file.
-The description of the characteristics of the SAIL model in 2.1 is deficient and formula (1) is not clear enough.
Response:
In the new manuscript, The description of the characteristics of the SAIL model were revised and added.
The SAIL model is a canopy radiative transfer model developed by Verhoef through extending the Suits model to allow for the variations of leaf angles. The model assumes that the canopy is consist of small absorbing and scattering elements, with known optical properties, distributed randomly in horizontal layers and with a known angular distribution [30].It permits to calculate the bidirectional reflectance factor and FPAR of plant canopies by solving the scattering and absorption of four upward/ downward radiative fluxes [31,32]. The four-stream differential equations form the basis of the SAIL models. These equations describe the contribution of the vegetation components illuminated by three fluxes () to the flux toward the observer () and the attenuation of this flux along the path:
where, is direct solar irradiance,is diffuse downward irradiance, is diffuse upward irradiance,is radiance in the direction of view. The detailed calculated can be found in reference [22].
The SAIL model requires only a few parameters including leaf reflectance and transmittance, leaf area index (LAI), leaf inclination distribution function (LIDF), the observational geometry and background reflectance. Evaluations of the SAIL model have shown generally adequate agreement with observations. Presently, the SAIL model has been widely used in remote sensing research for investigating and describing absorption and reflectance properties of vegetation canopies [33, 34]. In the SAIL model, all components in the canopy are randomly mixed in a single vegetation layer. As a result, the output FPAR is the canopy total FPAR, and rather than the FPAR of each component. As well known, in the SAIL model the canopy is assumed as a number of layers which are made up of small flat leaves in the SAIL model. The configuration provides a means to separate the FPAR for different components. If a canopy is divided into multiple layers and each layer only contains one component during simulation, the FPAR in each layer can be calculated, respectively. Thus, the radiation absorbed by green leaves can be separated from the total FPAR. Based on this point, the SAIL model was reconfigured in our study. In the revised model, FPAR for each layer can be calculated from the three flux streams as the balance between energy into and out of that layer. For example, FPAR the first layer can be described as following
where, is the upward diffuse flux;
is the downward diffuse flux;
is the downward specular flux;
The number in bracket denotes the layer number (e.g.1 is the top layer, 2 is the
second layer).
- Figure 1 is not referred to or commented on in the text. Also, it is not clear why two figures are needed and why “branch reflectance” is repeated twice.
Response:
In the manuscript, we simulated broadleaf and needle leaf trees, so there are two branch reflectance in Figure1.
In the new manuscript, the text was revised.
“In this study, broadleaf and coniferous forests were considered. The forest spectral data comes from the Superior National Forest study. Six tree species including red alder, white oak, aspen, western hemlock, western juniper and sitka spruce were selected. The leaf and branch spectra of broadleaf tree species and needle leaf tree species were averaged respectively (Figure 1). These averaged spectra inputted into the SAIL model to simulate reflectance and FPAR for broadleaf and needle leaf canopies. “
- All the figures are barely legible.
Response:
In the new manuscript, all the figures were reprocessed.
- Figure 2 has a different caption than the others “Percent PAV LAI” vs “Proportion of PAV area”. Furthermore, there is an extra image (Plant Area Index vs Total FPAR) which is not present in figure 3 and is not discussed. Why is total FPAR not sensitive to LAI? Check the spelling of the X-axis label (Plant not Pland).
Response:
Thanks.
In the new manuscript, Figure2 and Figure 3 were reprocessed. The extra image was removed .
Figure 2. FPAR of broadleaf deciduous forests canopies from the SAIL model
Figure 3. FPAR of evergreen coniferous forests canopies from the SAIL model
The total FPAR is sensitive to LAI. In the manuscript, there is a sentence “The total FPAR and green FPAR increased as canopy PAI was increasing.”
According to the SAIL model results, vegetation indices have more significant variations when plant area index varies, while the variation of proportion of leaf area produces fewer variations in vegetation indices. In the manuscript, there is no statistical analysis, but a comparison in changes of vegetation indices is conducted when proportion of leaf area and plant area index vary.
- In 3.2 you write about significant correlations, but no statistical analysis is reported.
Response:
In the new manuscript, we rewrote the 3.2. Statistical analysis was added.
“Figure 6 demonstrates the variations of canopy FPAR (including FPARPAV, FPARNPV and the total FPAR) with four vegetation indices for broadleaf deciduous forests. The total FPAR, FPARPAV and FPARNPV had different change patterns with vegetation indices. The total FPAR and FPARPAV increased with vegetation indices regardless of the proportion of leaf area. For FPARNPV, its variations with vegetation indices were more complex than the total FPAR and FPARPAV. With the increase of vegetation indices, FPARPAV increased initially (PAI <0.2) and thereafter decreased (PAI >0.2). Furthermore, the trajectory of FPAR-vegetation index varied with the proportion of leaf area in 2-D space composed of FPAR and vegetation index. For a given value of anyone of vegetation indices, FPARPAV values increased with the proportion of leaf area in canopies, while the total FPAR and FPARNPV decreased. Correlation coefficients between the total FPAR, FPARPAV and each vegetation index were high and can pass the test at the 0.01 significance level, which indicated there were significant linear positive correlations between them. However, correlation coefficients between FPARNPV and four vegetation indices were low(r<0.12). Par-ticularly, the correlation coefficient between FPARNPV and EVI2 was only 0.019. NDVI had closer relationship with the total FAPR than other three vegetation indices. EVI demonstrated better consistence with FPARPAV. When the proportions of leaf area rose, the correlation coefficients between FPAR and four vegetation indices increased. For example, the correlation coefficient between FPARPAV and NDVI, EVI, EVI2, SAVI de-creased by 0.03,0.015,0.017,0.022 when the proportions of leaf area in canopies varied from 0.5 to 0.98.
For evergreen coniferous forests, the relationships between the canopy FPAR (including FPARPAV, FPARNPV and total FPAR) and four vegetation indices were similar with those of broadleaf deciduous forests (Figure 7). The significant linear relationship can be found between canopy FPAR (including the total FPAR, FPARPAV) and each vegetation index, while the relationship between FPARNPV and each vegetation index was nonlinear. For evergreen coniferous, the total FPAR and FPARPAV had the closest relation with NDVI(r=0.993), EVI(r=0.99), respectively, but correlation coefficients FPARNPV between and four vegetation indices were less than 0.07.Withthe increase of the proportion of leaf are in canopies, correlation coefficients between FPARPAV and four vegetation indices increased.”
- Figure 6. Axes ranges should be the same.
Response
Thanks. In the new manuscript, Figure6 were reprocessed
- In the discussion more results of simulations are reported. Again, at the beginning of 4.2, you write about correlations without any statistical analysis. Furthermore, no statistical analysis is reported for your interpolated formulas.
Response:
In the new manuscript, we analyzed the errors of the two formulas
The errros were small when FPARPAV was estimated through the model.For deciduous broadleaf forests, the mean error of FPARPAV was 0.0086,and the error rate was 1.99%,while the mean error was 0.026 and the error rate was 3.76% for evergreen coniferous forests. It seems that the inversion model for deciduous broadleaf forests can produce better results than the model for evergreen coniferous forests.

Reviewer 3 Report
A unique paper on the capabilities of photosynthesis. In particular, the focus on the crown structure is highly original.
Consider the following points.
In this study, alder, aspen, oak, hemlock, and spruce were targeted. However, each of these tree species has different physiological characteristics, such as shade tolerance, and photosynthetic ability. Please provide supplementary explanations about the correlation between the physiological characteristics (especially shade tolerance) of each of these trees and the results of this study.
Author Response
Thank you sincerely for your advice. You advice improves the quality of the study, and furthermore, provides some ideas for further studying. Thank you again!!
In this study, our objective are to explore the relation between the green FPAR and vegetation indices, and furthermore to inverse green FPAR using remote sensing. We used the spectral data of six different tree species. Samples of components in canopies were collected and their spectral data were measured in the laboratory using a spectrometer. In this study, we simulated canopies FPAR and optical properties based on the SAIL model. Presently, the SAIL model the physiological characteristics were not considered in the SAIL model.
Each of these tree species has different physiological characteristics, and they influence the light absorption of the canopy. When these characteristics change, optical properties of canopies have different performance. It is very necessary to analyzing the influence of physiological characteristics on reflectance of canopies during the application of remote sensing data, particularly for vegetation parameters inversion. In the study, we analyzed the influence of the canopy structure on optical properties and light absorption capacities through simulation and constructed the inversion model of green FPAR. This is the first step, and environmental factors were not considered. Following this, we will conduct some field experiments, and these physiological characteristics will be considered and analyzed.
Thank you sincerely for the advice.

Round 2
Reviewer 2 Report
The manuscript was improved and many concerns were fixed. However I still have some suggestions:
1) Statistics should be improved. Add one or more tables with all the correlation coefficient. Moreover, you should detail which method was used for assessing significance level.
2) Specify the interpolation method used to obtain the formulas in lines 502 503.
3) I still think that what you report in paragraph 4 are mainly results rather than discussion. I suggest merging it with paragraph 3 (which you could title Results and discussion).
4) It must be emphasised in the conclusion that the results of the model have not been validated.
Author Response
Thank you sincerely for your advice. Your advice is helpful for improving the quality of the study. Thank you again!!
Question 1) Statistics should be improved. Add one or more tables with all the correlation coefficient. Moreover, you should detail which method was used for assessing significance level.
Response:
The Pearson correlation coefficients between FPAR and vegetation indices were calculated to measure the linear correlation between the two, respectively. The significance level was assessed based on Pearson correlation critical values table. In the new manuscript, statistics were described as follows:
Pearson correlation coefficients between the total FPAR, FPARPAV and each vegetation index were calculated, and they were high and can pass the test at the 0.01 significance level according to Pearson correlation critical values table.
Question 2) Specify the interpolation method used to obtain the formulas in lines 502 503.
Response:
In the study, the linear regression method was used to construct the formulas. The paper was revised through adding a sentence:
Through the linear regression method, the formulas between FPARPAV and EVI for deciduous broadleaf forests and evergreen coniferous forests were constructed. Their FPARPAV can be represented as respectively.
Question 3) I still think that what you report in paragraph 4 are mainly results rather than discussion. I suggest merging it with paragraph 3 (which you could title Results and discussion).
Response:
Thanks. The results and discussion were merged in the new manuscript.
Question 4) It must be emphasized in the conclusion that the results of the model have not been validated.
Response:
Thanks. In the new manuscript, a sentence was added: For two typical forests, two formulas to calculate green FPAR are constructed, respectively. It should be noted that the results of the model have not been validated due to the lack of field green FPAR.
